# Halide perovskite memristors as flexible and reconfigurable physical unclonable functions

Rohit Abraham John [1,4], Nimesh Shah[2,4], Sujaya Kumar Vishwanath[1,4], Si En Ng[1,4], Benny Febriansyah[3,4], Metikoti Jagadeeswararao[3], Chip-Hong Chang [2], Arindam Basu [2✉] & Nripan Mathews [1,3✉]

Physical Unclonable Functions (PUFs) address the inherent limitations of conventional hardware security solutions in edge-computing devices. Despite impressive demonstrations with silicon circuits and crossbars of oxide memristors, realizing efficient roots of trust for resource-constrained hardware remains a significant challenge. Hybrid organic electronic materials with a rich reservoir of exotic switching physics offer an attractive, inexpensive alternative to design efficient cryptographic hardware, but have not been investigated till date. Here, we report a breakthrough security primitive exploiting the switching physics of one dimensional halide perovskite memristors as excellent sources of entropy for secure key generation and device authentication. Measurements of a prototypical 1 kb propyl pyridinium lead iodide (PrPyr[PbI$_3$]) weak memristor PUF with a differential write-back strategy reveals near ideal uniformity, uniqueness and reliability without additional area and power overheads. Cycle-to-cycle write variability enables reconfigurability, while in-memory computing empowers a strong recurrent PUF construction to thwart machine learning attacks.

[1] School of Materials Science and Engineering, Nanyang Technological University, Singapore, Singapore. [2] School of Electrical and Electronic Engineering, Nanyang Technological University, Singapore, Singapore. [3] Energy Research Institute @ NTU (ERI@N), Nanyang Technological University, Singapore, Singapore. [4]These authors contributed equally: Rohit Abraham John, Nimesh Shah, Sujaya Kumar Vishwanath, Si En Ng, Benny Febriansyah. ✉email: arindam.basu@ntu.edu.sg; nripan@ntu.edu.sg

With the emergence of the edge computing paradigm within the Internet of Things (IoT) framework, we increasingly rely on connected network devices to manage vast volumes of digital confidential information and perform security-sensitive tasks on a daily basis. These ubiquitous electronic devices have seamless access to secure information including but not limited to biometrics, e-mail passwords, credit card information, and authentication tokens for financial transactions. This has made edge computing devices an attractive target and gateway to cyber-terror hacks, posing a large threat to information security[1]. The inability of software-oriented security alone to combat the increasing threats of cyberspace has called for the development of hardware cryptographic solutions[2]. Conventional hardware cryptographic solutions typically rely on non-volatile memory arrays for storing a secret key that is used by different cryptographic algorithms. The security of the key is of utmost importance since it is the only unknown variable that an attacker needs to recover or break any cryptosystem according to Kerckhoff's principle and Shannon's maxim. However, maintaining the key as a secret over the lifetime of the device is challenging due to the non-volatile, digital nature of the memory which retains the information even when powered down. Moreover, it is also difficult to bind the key to a specific device since copies of the key can be made[3].

Physical Unclonable Functions (PUFs) are among the most promising hardware cryptographic primitives capable of generating random, unique secret bit-strings on the fly[3,4]. Their feasibility stems from the unique signatures induced in a component's physical property via stochastic variations in the manufacturing process, thus making cloning impossible even by the original manufacturer[5,6]. Mathematically, a PUF can be appropriately modeled as a function that maps inputs/challenges ($c$) to outputs/responses (r), i.e., [$r = f(c)$], generating a secret challenge-response pair (CRP) space that is known only to a verifier[3]. They can be broadly classified as weak or strong PUFs[1,4] depending on whether the number of available challenge-response pairs (CRPs) are linearly or supra-linearly related to the number of physical random elements in the circuit. Weak PUFs are generally used for chip identification while strong PUFs are used for applications like authentication requiring the generation of many keys. State-of-the-art PUF designs are mainly based on Complementary Metal Oxide Semiconductor (CMOS) technology—e.g., gate delays in multiplexers (Arbiter PUF)[7], frequencies of electronic oscillators (Ring Oscillator PUF or ROPUF)[8] and randomness of the initial state of memory (SRAM PUF)[9]. However, the emergence of nanoelectronics and breakthroughs in new semiconducting materials and device configurations have provided computer engineers with entirely new state variables such as electron spin and resistance to represent information[10–12]. These technologies, while offering ultra-efficient and unique circuit opportunities for PUF design, could also potentially trigger new modalities for attack and render existing defenses unacceptable. Therefore, there is a critical need to evaluate robust "roots of trust" in the context of new nanoscale devices[2,13,14].

Memristor[15] is one such technology that is envisioned to push the future of computing beyond Moore's law by enabling new neuromorphic concepts such as in-memory computing[16–18]. Unlike traditional non-volatile memory (NVM)-based secret key storage, here the digital bits of the key could be extracted on the fly by utilizing the entropy existent within the variations in the high or low resistance states—this makes it extremely difficult to read out the key directly as is the case with the traditional method of the digital key stored in an NVM. In contrast to the conventional silicon PUFs that solely rely on manufacturing process variations, preliminary investigations on memristors have revealed unique PUF designs that harness the intrinsic stochasticity in their physical mechanisms as sources of entropy—e.g., resistance variation between operational cycles and probabilistic switching behaviors[19]. However, such investigations have to date been limited to only oxide memristive materials such as $TiO_x$, $TaO_x$, and $HfO_x$[20–23].

Novel unconventional electronic materials[24] such as organic semiconductors and halide perovskites provide a unique opportunity to design highly secure PUFs with a large entropy but have not been investigated to date. These materials provide a facile and fertile playground for tailoring optical, electronic, and electrochemical properties through a combination of rational chemical synthesis, functionalization, fabrication processes, and choice of appropriate interfacial layers, thus enabling systems with high degrees of freedom[25]. All these phenomena can be viewed as novel sources of entropy to design highly secure PUFs but are hitherto undemonstrated. The ability of these new material systems to be processed at low temperatures through inexpensive solution-based manufacturing techniques opens up the possibilities of cost-effective printed devices. With such printed components poised to be integrated into mass-manufactured products and low-cost electronics, printable hardware security systems would be of significance. Such printed and flexible sensors, and electronics built on a new generation of advanced materials are expected to propel innovations in smart medical technology, automotive, manufacturing, IoT, and consumer electronics[26]. Despite remarkable progress in the realization of large-scale passive and active circuitry with organic electronics[27,28], the hardware security aspect in these systems is thus far unaddressed.

An ideal electronic PUF material should possess multidimensional entropy—co-existence of intimately coupled multiple charge transport properties that would act as a black box to not only the attacker but also to the original manufacturer. Hybrid organic-inorganic memristors based on $ABX_3$ halide perovskites (where $A = Cs$, $CH_3NH_3^+$, $HC(NH_2)_2^+$; $B = Pb^{2+}$, $Sn^{2+}$, $Ge^{2+}$, $Cu^{2+}$; and $X = I^-$, $Br^-$, $Cl^-$) is one such material platform that possesses very large degrees of freedom or state variables, making them ideal candidates for next-generation digital fingerprints/PUFs (Fig. 1). They encompass a wide variety of switching physics capable of acting as proxies for memristive switching, including but not limited to modulatable ion migration, self-doping, electrochemical metallization, and localized interfacial doping, as detailed in Supplementary Note 1. Unique to halide perovskites (HPs), is the co-existence and coupling of ionic and electronic components of current and capacitance, resulting in demonstrations of switchable majority carrier concentrations, giant dielectric constants, intrinsic localized doping, and above bandgap photo-voltages[29–32].

Here, we hypothesize that embracing these intrinsically coupled sources of entropy in halide perovskite memristor PUFs (HP memPUFs) in combination with advanced bit-stabilization techniques will be advantageous to create robust hardware security primitives for secure key generation and device authentication (Fig. 1). While some works have explored luminescence-based anti-counterfeiting and information storage using perovskites[33–36], they fall short of providing rigorous proof for the quality of stochasticity and encoding capacity, cannot produce a multitude of keys required for authentication applications; neither can their keys be reconfigured in case of information leakage. We experimentally demonstrate reconfigurable weak PUFs or physically obfuscated keys (POKs) with a 1 kb array of HP memristors. This is the very first implementation of an HP memPUF and also the largest ever implemented HP memristor array to date to the best of our knowledge. The secret keys are generated from the measured resistance distribution of HP memPUFs as the source of entropy. To enhance the robustness and reliability of the responses obtained from an instance upon

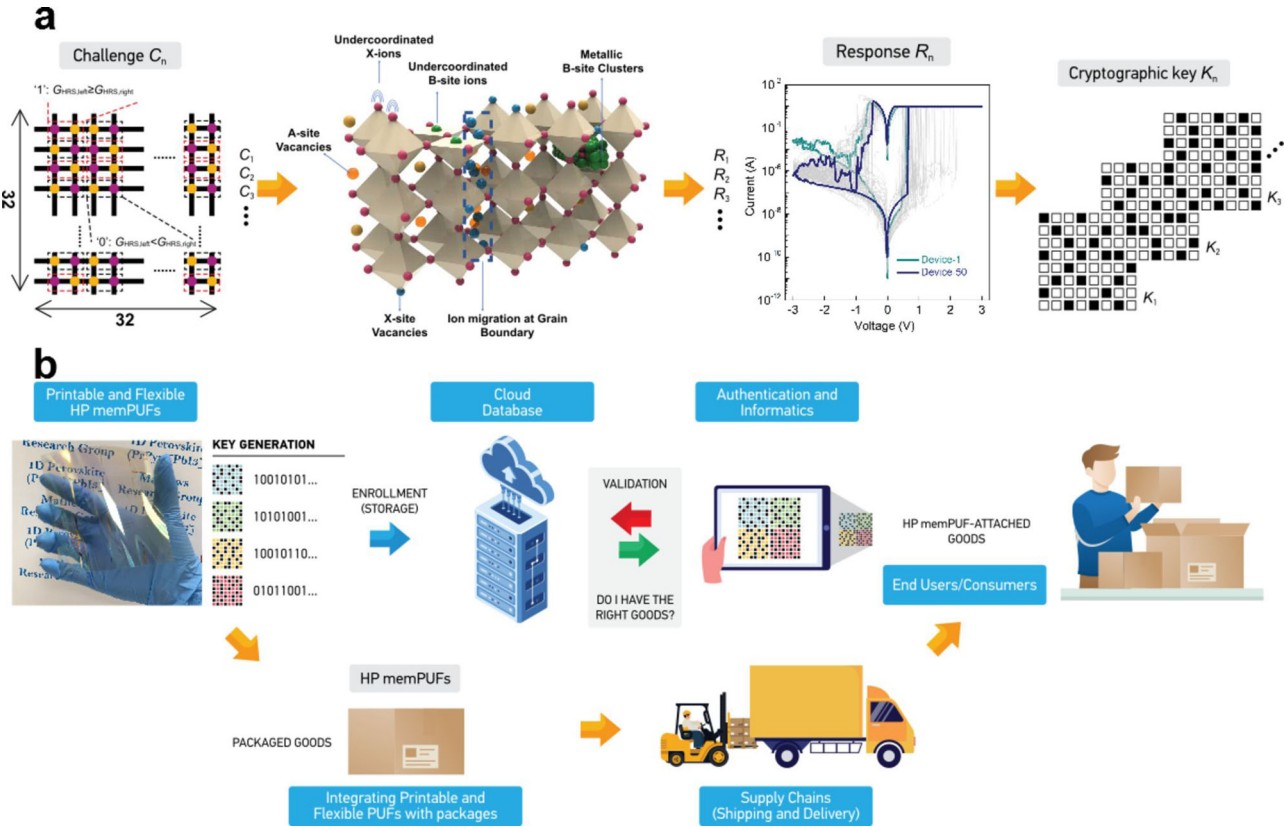

**Fig. 1 Halide perovskite memristor PUFs (HP memPUFs). a** HPs possess a rich reservoir of intimately coupled charge transport properties that could serve as sources of entropy to design new kinds of PUFs. Examples include electrochemical metallization reactions, formation of metallic Pb clusters, vacancy-driven valence change mechanisms, ion migration, grain boundary effects, and so on. Here, stochasticity in the high resistance state of HP memPUF cells in the crossbar provides parametric support for the generation of unique cryptographic keys. Design flow: A differential readout strategy is adopted to read the resistance of the memory cells→ coexistence of a multitude of switching physics in halide perovskites enables high stochasticity of the resistance states→ the differences in the I–V curves of the memristor cells reflect this stochasticity→ cryptographic keys are generated from the difference in the high resistance state of these memristors. **b** Concept schematic of product authentication. The flexible HP memPUFs developed can be attached to the packaging of goods or products by the original manufacturer. Each individual product would have a unique PUF ID. End users (e.g., consumers or other entities along the supply chain) can validate the product and certify the provenance by accessing the enrolled keys in a secure cloud database. In addition, the HP memPUF array could be used to store other useful information such as product expiration date, supply chain path, etc relevant to the specific product.

repeated querying, we adopt a write-back strategy exploiting the write mode of the HP memristors. We demonstrate this to be more effective than conventional improvement techniques such as temporal majority voting and fuzzy extractors[4] that provide low gain and require significant additional power and area overheads, respectively. To combat machine learning algorithms that build models of the randomness by eavesdropping on several hundreds of CRPs, we utilize concepts of recurrence (common in neural networks) to provide a hardware frugal solution in comparison to increasing the size of the crossbar array[21,37]. In short, we demonstrate (i) both weak and strong PUF modes; (ii) write-back assisted near-perfect uniqueness, reliability, and randomness; (iii) reconfigurability to refresh keys when necessary; (iv) as well as a recurrent scheme of response generation that enhances the security of small crossbar arrays against machine learning attacks. Compared to state-of-the-art CMOS and oxide memristor PUFs[22,23,38–43], our devices are flexible and can be facilely printed on large areas while providing comparable performance. With respect to other recently reported oxide memristor PUFs[22,23,44], our devices pass all the 15 randomness tests from the National Institute of Science and Technology (NIST) and thwarts machine learning attacks on the strong PUF mode by utilizing recurrence.

## Results

To achieve the purpose, we utilize a pyridinium-based HP material featuring molecular one-dimensional (1-D) lead-iodide lattices, namely propyl pyridinium lead iodide ($PrPyr[PbI_3]$). Higher electronic confinement, presence of charge trap states, and halide migration along the 1-D chain result in superior resistive switching when compared to their 3D and 2D counterparts[45], as detailed in Fig. 2, Supplementary Note 1, Supplementary Fig. 1, Supplementary Table 1. We hypothesize that all of these mechanisms would contribute to the stochasticity in conductive path formation in PrPyr $[PbI_3]$ memristors as a proxy to create PUFs that generate cryptographic keys with multiple challenge-response pairs.

Figure 2a shows the crystallographic structure of $PrPyr[PbI_3]$ along the "a" axis. 200 nm thick $PrPyr[PbI_3]$ is sandwiched between poly(methylmethacrylate) (PMMA, 20 nm) and poly(3,4-ethylenedioxythiophene): poly(styrene-sulfonate) (PEDOT:PSS, 50 nm) layers with Ag (~100 nm thick, square $100\,\mu m \times 100\,\mu m$) and indium tin oxide (ITO) ($10\,\Omega$/sq) as the top and bottom electrodes, respectively (Fig. 2b, c). To benchmark the memristive switching characteristics of HP memristors, we first fabricate and analyze dot-array structures of Ag/PMMA/PrPyr[PbI3]/PEDOT:PSS/ITO on a flexible polyethylene terephthalate (PET) substrate (Fig. 2d). Figure 2e depicts representative resistive switching operations of PrPyr

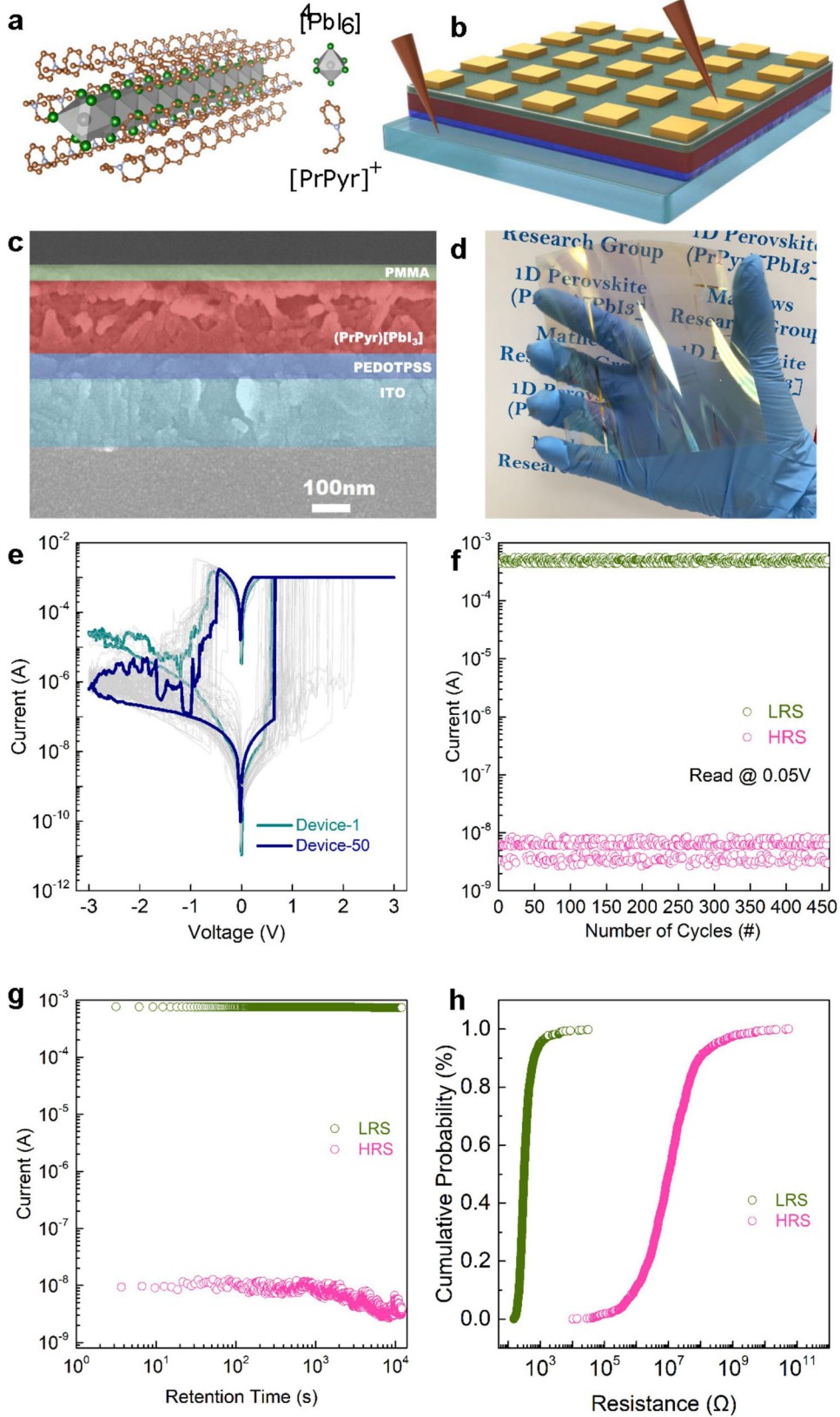

[PbI$_3$] memristors, measured across 50 devices (0 V → +3 V → 0 V → −3 V → 0 V, step size = 20 mV). All the devices display excellent bipolar resistive switching behavior with a high on-off ratio of 10$^5$, ensuring a wide programming margin that enhances the reliability of PUFs. Endurance testing performed in the alternating current (AC) mode reveals excellent stability of the switching

process (Fig. 2f). The cycle-to-cycle variations are exploited for reconfiguration, as discussed later. The devices maintain a high on-off ratio of >10$^5$ over 450 switching cycles and exhibit a retention time of ~10$^4$ s (Fig. 2g), reiterating the stability of the switching process. All these features represent a significant advance over previous reports of halide perovskite memristors where the

**Fig. 2 1-D Halide perovskite (HP) memristors. a** Single-crystal X-ray structure of 1-D (PrPyr)[PbI₃]. Gray, green, brown and light blue spheroids correspond to Pb, I, C, and N atoms, respectively. H atoms are omitted for clarity. **b** Schematic and **c** cross-sectional scanning electron microscopy (SEM) image of the memristor device structure- Ag/PMMA/PrPyr[PbI₃]/PEDOT:PSS/ITO. **d** Photograph of the large-area memristor array fabricated on a flexible PET substrate. **e** Representative DC I–V switching characteristics of HP memristors. The voltage sweeps are applied across the top Ag electrode with the bottom ITO electrode grounded. The devices switch (set) seamlessly from its high resistance state (HRS) to low resistance state (LRS) at ~+0.84 V (average of 50 devices) when the voltage is swept from $0\,V \rightarrow +3\,V \rightarrow 0\,V$. During set, a compliance current of 1 mA is applied to prevent the devices from hard breakdown. On reversing the polarity- $0\,V \rightarrow -3\,V \rightarrow 0\,V$, the devices are reset back to their initial HRS at ~−0.51 V (average of 50 devices). **f** AC endurance results reflect the excellent stability of the switching process for 450 cycles. Write voltage pulses are applied to the top electrode ($V_{set} = +1.2\,V/10\,ms$ and $V_{reset} = -1.5\,V/10\,ms$), followed by read pulses ($V_{read} = 0.05\,V$). A read voltage of 0.05 V is chosen in order to avoid unintentional switching of the memory devices during the measurements. **g** Retention and **h** cumulative probability distribution plot of the LRS (green) and HRS (pink) across 1024 devices.

switching performance markedly deteriorated on flexible substrates[46,47], highlighting the robustness of 1-D PrPyr[PbI₃] memristors. Cumulative probability distribution plot (CDF) of the LRS (green) and HRS (pink) measured across 1024 devices indicate larger variations among devices for the HRS and hence we choose HRS as the entropy source for the PUF design (Fig. 2h, Supplementary Note 2, Supplementary Fig. 2).

**Weak PUF bit generation and characterization of security metrics**. To implement the HP memPUF (weak or strong), all devices are initially set to their on-state (low resistance state/LRS) via a forming process, performed row by row. Next, all the cells are reset to their off-state (high resistance state/HRS) and the variation that occurs in the reset process is used as the entropy source (Supplementary Note 2, Supplementary Fig. 2). The CDF of the resistances in HRS display >3.5× higher coefficient of variation than that in the LRS, justifying our choice of HRS as the entropy source. Figure 3a shows the flowchart and circuit to generate response bits in the weak PUF mode of the HP memristor array. Further details are provided in Supplementary Note 3. Consecutive 1-D HP device pairs in one row are considered as a different unit cell (2R cell). This differential design we adopt helps to cancel out first-order environmental dependencies. From our $32 \times 32$ arrays of HP memristors, we obtain 512 independent pairs and therefore uncorrelated 512 bits. Note that, unlike some previous proposals that generate bits based on memory writes, our method only performs write once and hence can be used to uniquely identify a device for authentication without suffering from the limited write endurance of 1-D HP resistive random access memory (RRAM) devices[48,49]. To further reduce the temporal fluctuations in resistance/read-out noise, the sensing margin is increased by setting a pair of devices to their complementary states i.e., if device A's current is greater than that of device B's, device A is set to LRS while device B is kept in HRS. This "write-back" process[50,51] (Fig. 3a, Supplementary Note 3) ensures excellent stability of the response bit against undesirable temporal, voltage, and temperature fluctuations (as discussed later) (Fig. 3b).

The PUF produces a device-unique key by transforming the random analog conductance values to binary digits. Here, we present a 1024 element ($A = B = 32$) 1-D HP memPUF, the largest HP memristor array reported so far. Figure 3c shows an example of an analog resistance map, as well as the digital bits generated from this array using the process described earlier. To benchmark and assess the security strength of an HP memPUF instance, we systematically evaluate important metrics such as uniformity (UF), uniqueness (UQ), and reliability (REL)[3] (Supplementary Note 4).

We begin by analyzing the analog distribution of the 1st HRS states across the 1 kb array without any write-back step. UQ and REL are first analyzed by calculating the inter-HDs and intra-HDs over the 1 kb 1-D HP memPUF array (Fig. 3d). Intra-HD is measured for temporal changes over 100 cycles for each of the 1024 devices in the array. The mean intra-HD obtained is

2.33%, which shows good reliability over temporal variations without any kind of post-processing or error correction. To calculate inter-HD or uniqueness (UQ), we create eight groups of 128 devices each by partitioning the array evenly. For each group, we generate 64-bit keys out of 128 analog values, and we calculate the HD for $\binom{8}{2} = 28$ pairs of devices. We obtain a mean value of 48.3% with a 6.8% standard deviation, showing excellent separation between inter and intra-HD. These results for the facilely solution-processed HP memPUF array compare favorably with the state-of-the-art lithographically-patterned oxide memristors with complex fabrication steps, highlighting the utility of our approach[22]. To estimate the distribution of uniformity (UF), we sequentially create 64 groups of 8 bits each and calculate UF for each group. A mean UF of 49.02% is obtained for the 512 bits, which shows the generated bitstream has minimal bias. The histogram of the uniformity results presented in Supplementary Note 4, Supplementary Fig. 3b and the ability to pass all 15 National Institute of Standards and Technology (NIST) tests (see below and Supplementary Note 10, Supplementary Table 2) confirms the comparable or better quality of random bit stream generated from our devices compared to other designs[22,52–54] (Supplementary Note 11, Supplementary Table 3).

As mentioned earlier, here we exploit the memory primitive and adopt a write-back step[51] that allows us to implement highly reliable PUFs[51] (Fig. 3a). In comparison to conventional approaches such as forward or reverse fuzzy extractors that use error correction codes (ECC) with helper data to regenerate the secret keys within noisy channels, our approach saves non-trivial costs of time, area, and power overheads. Moreover, such multiple exposures of helper data for a single PUF response could also lead to min-entropy leakage, which our write strategy avoids. Figure 3b plots the distribution of the resistances of the 1-D HP memPUF after write-back displaying an increased sensing margin. Figure 3c shows the analog color map of HRS and the digitized checkerboard pattern after write-back. The checkerboard pattern illustrates the random distribution of 1's and 0's of the weak PUF key. We next evaluate the effect of write-back on the PUF metrics. Since the two memristor cells after the write-back operation are in complementary states, the BER of regenerated response bit is 0% (intra-HD in Fig. 3d) for temporal changes compared to 2.33% without write-back. The mean UQ and UF values after write-back (Supplementary Note 4) are 48.1% and 48.44%, respectively suggesting no adverse effect of write-back on these metrics. The randomness of the response bitstream was further tested using the NIST test suite (Supplementary Note 10, Supplementary Table 2), resulting in a pass for all the fifteen tests, vide infra.

To further assess the effect of write-back on the metrics of our PUF instances in harsh operating conditions, we next evaluate inter and intra-HD metrics for 16 devices (representative of 1 kb)

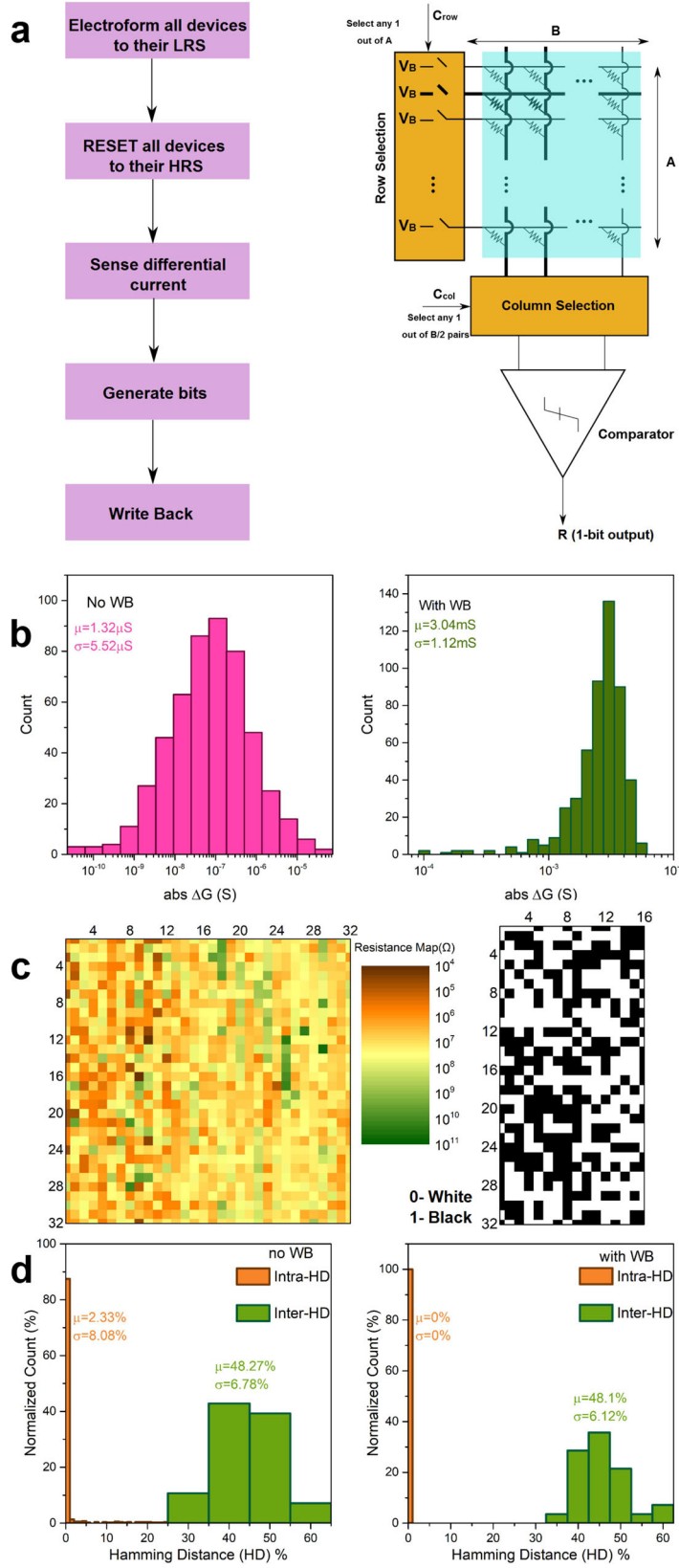

**Fig. 3 1-D Halide Perovskite (HP) Weak memristor Physical Unclonable Function (memPUF). a** Flowchart of HP memPUF bit generation (left). Circuit architecture of 1-D HP memPUF 32 × 32 crossbars, with differential sensing through a low-offset comparator (right). **b** Distribution of differential conductance before and after write-back. The magnitude of differential conductance shows a marked increase after write-back (WB) which results in improved reliability. **c** Analog map of HRS showing high randomness, and digital maps obtained after write-back. **d** Inter and Intra-HD before and after write-back for temporal changes show a reduction in bit error rate (BER) or intra-HD to 0% after write-back compared to 2.33% before write-back. Large separation between inter and intra-HD after write-back constitutes a good PUF that is identifiable and reliable.

against undesirable temporal, voltage, and temperature fluctuations (Supplementary Note 5, Supplementary Fig. 4). Temporal fluctuations are measured over 100 cycles, while a change in read voltage from 0.1 V to 0.15 V (i.e., 50%) is applied to the devices to represent noisy voltage channels. The effect of temperature fluctuations is analyzed for up to 85 °C. The mean intra-HD for temporal-only, voltage + temporal, and temperature + temporal fluctuations are 2.33%, 19.25%, and 63.42% at 60 °C, 63.71% at 85 °C, respectively. With write-back, the mean intra-HD reduces to 0% in all cases, reiterating the effectiveness of our write-back approach. An inter/intra-HD ratio of 2.6 without write-back and a value approaching infinity with write-back leaves sufficient tolerance for possible degradation of PUF UQ and REL without affecting false acceptance rates. Similarly, write-back can help in alleviating BER due to other perturbations such as bending or flexing of the 1-D HP memPUF as well. Furthermore, we demonstrate that the write-back technique reduces the BER to virtually 0% when compared to 0.21% using conventional techniques such as Temporal Majority Voting (TMV), reiterating the benefit of our approach (Supplementary Note 6, Supplementary Fig. 5).

**Reconfiguration.** Reconfiguration of the PUF key is often useful to support ownership change, change in privileges (e.g., software version downgrade), or preventing information leakage due to overuse[44,55]. To enable freshness of the cryptographic key, we envision cycle-to-cycle (C2C) write variability of the HP memristors to set/reset the cells into new states, i.e., 'physical reconfiguration'[44] of the HP memPUF (Fig. 4, Supplementary Note 7) as opposed to logical reconfiguration[56] which require additional circuits. With physical reconfigurability, the trustworthy party can now reconfigure all keys, making the previous keys inferred and documented by an attacker useless. Figure 4a shows the reconfiguration flowchart. Due to C2C variation, we obtain a new array of random conductances after every RESET to the HRS. The analog color map and checkerboard patterns in Fig. 4b, c illustrate the stark difference in distributions before and after reconfiguration. Figure 4d shows a histogram of the reconfiguration HD. The histogram is generated by calculating HD's of $\binom{10}{2} = 45$ pairs of bitstreams i.e., choosing a pair out of 10 cycles. Finally, the cross-correlation between the reconfigured keys exhibits a low value (Fig. 4e) with a mean value of 0.2 signifying good independence between the reconfigured keys.

**Strong PUF and modeling attack resistance.** We finally implement a strong PUF (Supplementary Notes 8 and 9) from our HP memPUFs by combining selected device currents efficiently via in-memory computing to get an exponential number of input-output pairs or CRPs. A chosen set of currents (according to row-challenge bits) along two selected columns are added (chosen by column challenge bits) using Kirchoff's Current Law (KCL) and comparing them with a sense amplifier to generate a response bit $b_R$[39] (Fig. 5a). This can be mathematically represented as:

$$b_R = 1 \text{ if } V_B \sum_{i=1}^{A} C_{\text{row,i}} G_{i,j} > V_B \sum_{i=1}^{A} C_{\text{row,i}} G_{i,k} \qquad (1)$$
$$= 0 \text{ otherwise}$$

where $C_{\text{row, i}}$ is the $i$-th bit in the row challenge, and $j$, $k$ denotes the column indices chosen by the column challenge. To avoid correlations between columns when selected arbitrarily, columns are paired with their immediate neighbor to avoid any information leakage (Supplementary Note 8) resulting in $B/2$ independent pairs. The number of bits in the challenge is $N_{\text{chal}} = A + \log_2(B/2)$

and the total number of CRPs are

$$C_{\max} = 2^A (B/2) \qquad (2)$$

which corresponds to $6.87 \times 10^{10}$ CRPs for $A = B = 32$ in our case. The quality of the response bits is evaluated for a total of $2 \times 10^7$ challenges and a REL of 94.04% is obtained. The randomness is further evaluated by taking blocks of $10^5$ bits or more (depending on the test requirement) and passing them through the NIST tests (Table 1). In comparison to other oxide RRAM PUFs[22,44], our proposed strong PUF passes all 15 tests proving the excellent quality of randomness of the CRPs.

One problem of the typical strong PUF design above is that it is extremely susceptible to machine learning attacks. If we consider a single column pair, the response is generated through a simple summation of a chosen set of conductances. If there are 'N' rows, a system of 'N' equations can be written to solve for 'N' variables. As shown in Fig. 5b, a variety of machine learners can be used to learn the CRP relationship in that case and achieve an accuracy of ~90%. We tackle this issue by using recurrence[57,58] to make the CRP relationship more obscure for machine learners to calculate. The idea involves generating 2 intermediate response bits using the vanilla strong PUF, XOR'ing these bits with the challenge to get an intermediate challenge, and then producing the final response bit using this intermediate challenge (Fig. 5c, Supplementary Note 8). The obfuscation comes from two sources: (1) XOR (2) two-column pair selections to get two intermediate response bits. Testing ML attacks on these new CRPs with the same training-testing split produces a maximum attack accuracy of 52% (Fig. 5d), far lower than the results without recurrence and close to the 50% accuracy of random guessing. Training with 10× increased number of samples, i.e., $10^6$ samples, reveals similar trends of better attack resilience with recurrence. For the worst-case assuming constant slope and extrapolating from $2^{20}$ CRPs, we estimate a collection of $2^{88}$ CRPs for successful impersonation (attack accuracy = 100%). Even if we assume an aggressive 10Mbps CRP collection rate, it will take $9.8 \times 10^{11}$ years for the attacker to collect $2^{88}$ CRPs to achieve close to 100% accuracy, demonstrating that our HP memPUFs are secure against the aforementioned ML attacks (Supplementary Note 8, Supplementary Fig. 6). This recurrence-based design can be further exploited to generate modeling-resistant keys from even smaller crossbars with as few as 4 columns that enable the operation of the current array even with many faulty devices. Hence, this could be considered as a universal approach to design ML-resistant strong PUFs using small to medium-sized device arrays made of unconventional electronic materials, where massive CMOS-like scaling is currently challenged.

## Discussion
We present stochasticity in the switching physics of one-dimensional halide perovskite (1-D HP) memristors as excellent sources of entropy to design physical-disorder-based security primitives for key generation and device authentication. This is a breakthrough advancement in the experimental realization of PUFs based on organic solution-processed materials and devices, the first of its kind to the best of our knowledge. Measurements from a prototype 1 kb 1-D HP memPUF array reveal near-ideal uniformity, uniqueness, and reliability of the primitive. The large on/off ratio of our 1-D HP memristor cells combined with the differential write-back strategy we adopt results in <1% intra-HD without additional area and power overheads when subjected to temporal noise, read voltage, and temperature variations, considerably better than other recent reports[22,23]. Since there are multiple sources of entropy within the HP memPUF, we expect

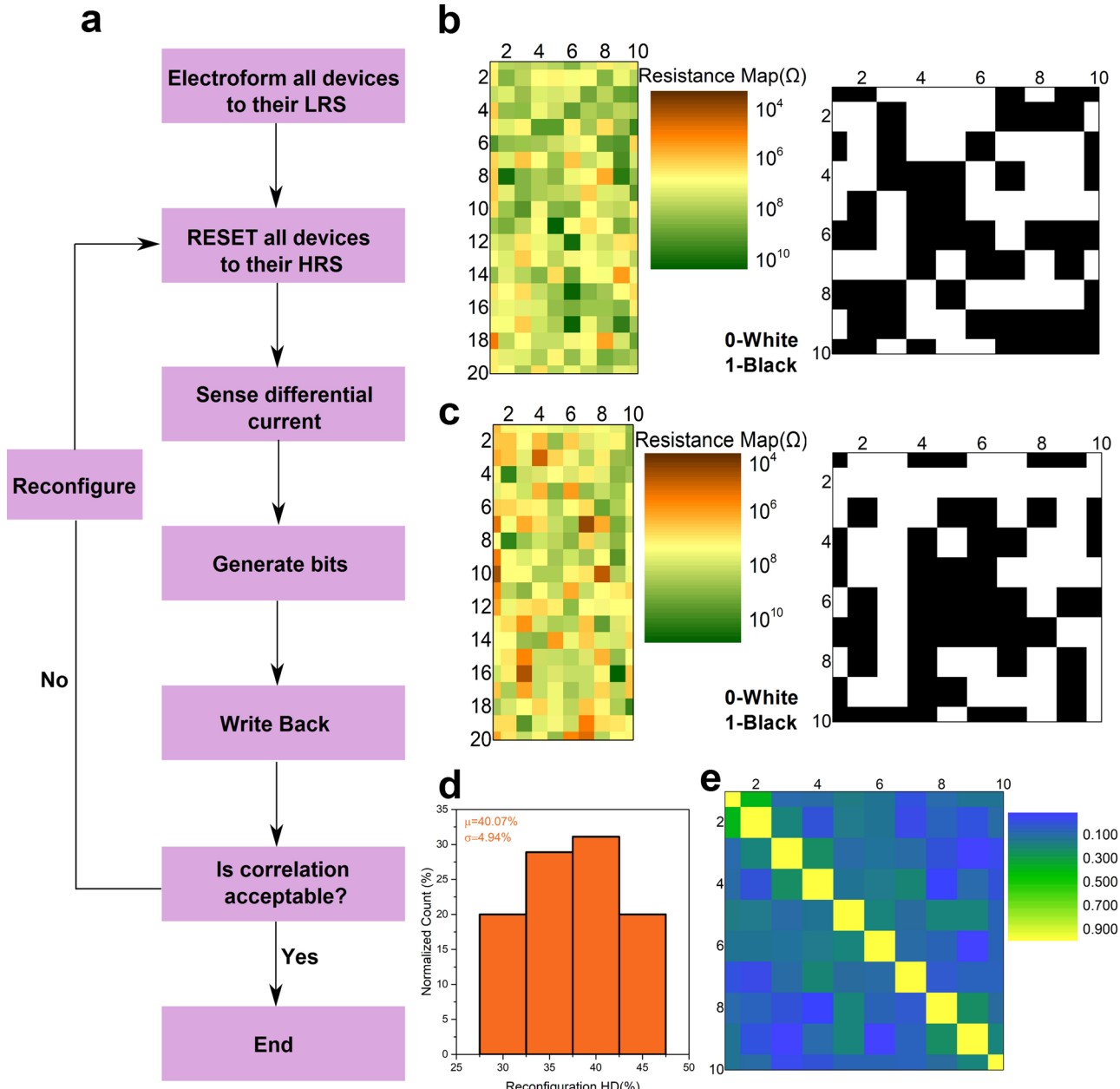

**Fig. 4 Reconfigurable halide perovskite (HP) memristor physical unclonable functions (memPUFs). a** Reconfiguration flowchart showing how reconfiguration is done exploiting cycle-to-cycle variability by resetting to HRS and running through the same protocol again every cycle to generate fresh keys. The PUF is repeatedly reconfigured as long as the correlation factor between the new key and the key from the previous cycle is not low enough. **b, c** Analog color map and digitized checkerboard before and after one cycle of reconfiguration showing the bits are different after reconfiguration. **d** 10 cycles of consecutive writes are done for reconfiguration. Histogram of reconfiguration HD obtained by calculating HD of $\binom{10}{2} = 45$ pairs of bit-streams, with a mean value of 40.06% showing good independence of the key bits across cycles. **e** Correlation matrix for 10 cycles of reconfiguration showing an average correlation of 0.2.

cloning to be impractical/improbable, and that an invasive attacker would have difficulty in extracting the secret key even with physical access to the devices since the physical functions would probably be destroyed upon attack. We further exploit the cycle-to-cycle write variability of the HP devices to enable reconfigurability of the weak PUF keys with ~40% inter-HD between the reconfigured keys. Finally, we demonstrate a strong PUF exploiting the in-memory computing feature of crossbars and use a universal recurrent PUF construction scheme to protect the PUF against modeling attacks without needing additional

crossbars[22]. With excellent power efficiency (Supplementary Note 9) and randomness, our 1 kb HP memPUF provides state-of-the-art performance compared to more mature oxide RRAM PUFs (Supplementary Note 11, Supplementary Table 3) in terms of increased stochasticity[22,59], higher native bit-stability than other non-CMOS reconfigurable PUFs[53] and oxide PUFs[23], and showcases unique features of reconfigurability and write-back compared to other CMOS PUFs such as arbiter and ROPUFs. Our approaches and conclusions are expected to be valid and future-proofed for large area crossbar implementations as well

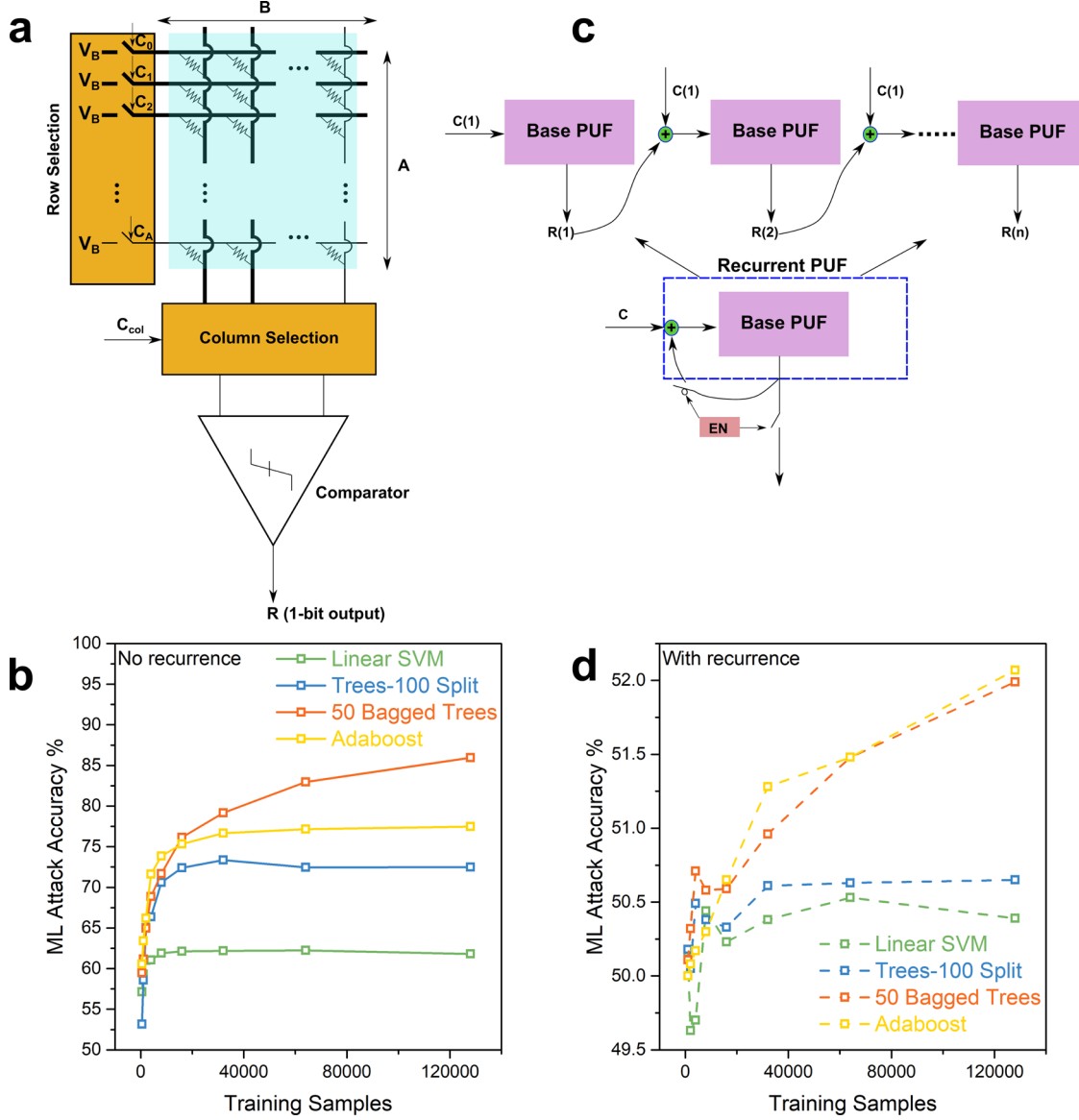

**Fig. 5 1-D HP strong memPUF resilient to machine learning attacks. a** Strong PUF architecture with a 1-bit response. Challenge consists of *A* bits for row selection (in this case, the first 3 bits are 1 and the last bit is 0) and $\log_2(B/2)$ bits for column pair selection (here the first bit is 1). Natural current summation along columns gives rise to an exponential number of CRPs. **b** Machine learning (ML) results for strong PUF without recurrence. Accuracies close to 90% reveal the HP memPUF to be highly susceptible to such attacks. **c** Conceptual diagram of a recurrent PUF configuration shown as an unrolled cascade of PUFs without recurrence. After "*n*" steps of recurrence, the EN signal is turned on to stop further recurrence and enable the output bit *R(n)* to be readout. **d** ML results for strong PUF with recurrence. Accuracies reduce to almost wild guess probability with recurrence compared to **c**, proving that recurrence improves resistance to ML attacks.

(Supplementary Note 12, Supplementary Fig. 7). This finds indispensable and immediate pertinence to the antipiracy of integrated circuits (ICs) and product authentication in supply chains.

## Methods

**(Propyl)pyridinium iodide (PrPyrI)**. Stoichiometric amounts of iodopropane were added dropwise to a solution of pyridine in dry acetonitrile. After stirring for 1–2 days (reaction completion was assessed by $^1$H NMR spectroscopy) at reflux, the reaction mixture was cooled down to room temperature before the solid product formed was isolated by filtration, thoroughly washed with diethyl ether, and brought to dryness under vacuum for 24 h. The iodide salt was then immediately transferred to the glovebox for further use.

**Growth of single crystals of hybrid 1-D material (PrPyr)[PbI₃]**. Stoichiometric amounts of PbI₂ and PrPyrI in DMF or DMSO, with a typical concentration of 0.1–0.25 M, were prepared in small vials before it is transferred into bigger vials

containing antisolvent acetone. 1-D perovskite single crystals thus formed by the vapor diffusion method, were then filtered and dried for X-ray crystallographic characterization and device fabrication.

**Device fabrication**. The memristor devices were fabricated on indium tin oxide (ITO) transparent conducting substrates. Initially, the ITO substrates were cleaned by sequential 15 min sonication in acetone, ethanol, and isopropanol (IPA). Subsequently, the substrates were dried using an $N_2$ gun and treated with ozone plasma for 15 mins. PEDOT:PSS (4086) was deposited by spin-coating at 4000 rpm for 80 s, followed by annealing at 110 °C for 10 mins. The PrPyrPbI₃ layer was next spin-coated at 4000 rpm for 30 s on top of PEDOT: PSS and baked on a hot plate for 5 mins at 100 °C. After cooling to room temperature, a thin poly (methyl methacrylate) (PMMA) film was deposited onto the perovskite film by spin-coating at 4000 rpm for 40 s . Finally, Ag electrodes were deposited with a shadow mask (100 μm × 100 μm) under vacuum ($10^{-6}$ torr) using thermal evaporation.

**Characterization**. Field emission scanning electron microscopy (FESEM, JEOL JSM-7600F) was employed to characterize the topographical and cross-sectional

**Table 1 National Institute of Standards and Technology (NIST) tests.**

| Test | $P$-value | No. of runs | Pass No. | Pass? |
|---|---|---|---|---|
| Frequency | 0.798 | 200 | 200 | Yes |
| Block frequency | 0.972 | 200 | 193 | Yes |
| Cumulative sums | 0.93 | 200 | 200 | Yes |
| Runs | 0.392 | 200 | 199 | Yes |
| Longest run | 0.575 | 200 | 197 | Yes |
| FFT | 0.036 | 200 | 196 | Yes |
| Serial | 0.72 | 200 | 196 | Yes |
| Linear complexity | 0.312 | 200 | 194 | Yes |
| Rank | 0.304 | 100 | 98 | Yes |
| Approximate entropy | 0.192 | 50 | 50 | Yes |
| Random excursions | 0.276 | 15 | 15 | Yes |
| Random excursions variant | 0.163 | 15 | 14 | Yes |
| Non-overlapping template | 0.658 | 50 | 50 | Yes |
| Overlapping template | 0.419 | 200 | 200 | Yes |
| Universal | 0.964 | 20 | 20 | Yes |

Results of NIST test for strong PUF construction. The test dataset consists of 20 million bits which correspond to 20 million CRPs. The responses pass all 15 tests.

images of the 1-D HP memristor. Glancing-angle X-ray diffraction measurements were conducted using a Bruker AXS D8 ADVANCE system with Cu Kα radiation ($\lambda = 1.5418$ Å). The XRD spectra were recorded with an incident angle of 5°, a step size of 0.05°, and a delay time of 1 s for each step. All electrical measurements were carried out using Keithley 4200 semiconductor parameter analyzer and a probe station.

## Data availability
The data that support the findings of this study are available from the corresponding authors upon reasonable request.

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

## Acknowledgements

The authors would like to acknowledge the funding from MOE Tier 2 grant MOE2018-T2-2-083, MOE Tier 1 grants RG87/16, NTU SUG, and NTU INT Funding CoE.

## Author contributions

R.A.J., A.B., and N.M. conceived the experiments. S.K.V., S.N., B.F., and M.J. fabricated the halide perovskite memristors under the supervision of N.M. R.A.J. conceived and set up the testing protocol. S.K.V., S.N., and R.A.J. characterized the halide perovskite memristors. N.S. performed all the PUF data analysis under the supervision of A.B. with advice from C.H.C. R.A.J., N.S., B.F., A.B., and N.M. wrote the manuscript with comments from all authors.

## Competing interests

The authors declare no competing interests.
