## [Peer Review File · Nature Communications]

REVIEWER COMMENTS

Reviewer #1 (Remarks to the Author):

The Authors report a PUF architecture which exploits the switching behavior of one dimensional halide perovskite memristors as a source of entropy.

The paper proposes measurements and claims that the proposed architecture shows the features to create a strong PUF resilient to machine learning attacks.

The work is well written and the Authors provide some interesting insight on a possible implementation of PUFs with perovskites. I have the following comments:

1) The Authors claim that the proposed 1D Halide Perovskite based memristors show endurance to the write cycles, however it is not clear whether the changes over the several cycles would actually modify the expected CRP, a discussion on this aspect would be useful in particular referring to the proposed write back scheme.

2) The Authors report that their PUF consists of a 32×32 array of RRAM dot-point devices as shown in Fig. 3a which they logically treat as a crossbar, leaving to future work to physically implement an actual crossbar, how does this assumption impact their analysis and the reported results?

3) When analyzing the ML attack resilience, the Authors report figure 5.1-b that their proposed approach based on recurrence lowers the accuracy of a ML attack to 52% in the worst case, however the trend of the curves does show a marked increase with the increase of training samples. This induces to infer that a higher number of samples could increase significantly the accuracy of the attack, the Authors should comment on this aspect.

Reviewer #2 (Remarks to the Author):

The study demonstrates a large scale implementation of security primitives exploiting the switching physics of 1D halide perovskite memristors for authentication that is reconfigurable. They implemented a 1 kb memory structure for a weak PUF demonstration.

The authors based their ideas on the co-existence and coupling of ionic and electronic components.

The authors claim their work is "the very first implementation of a HP memPUF" hence it is strange to also claim "the largest ever implemented HP memristor array till date" as if it is the first on HP then it is for sure the largest!

The material description, PUF analysis and outcome in my view are well argued and well presented. The type of material used in this paper gives me hope to see a much reduced cost in production of PUFs.

I am disappointed and it is unfortunate that this great work has to be rejected as it postpones one of the main aspects of their research product for future research. The authors states "However, the current 1D HP memPUF is still susceptible to large BERs with high temperature changes and is one of the major future directions of research."

The REL calculation for intra-HD is not assessed the temperature, based on what I found. Could you explain what do you mean by putting "We calculate both temporal and voltage-induced reliability" right after stating "response at any other temperature/voltage to determine if the responses are different (unreliable) or not (reliable)" in the Supp.? Are you linking temporal with temperature or are you implying the impact is transient?

The Reliability description in this paper is very unclear and vague in my view.

I suggest authors to at the very least demonstrate that devices respond on average the same way to changes in temperature.

REVIEWER COMMENTS

Reviewer #1 (Remarks to the Author):

The Authors report a PUF architecture which exploits the switching behavior of one dimensional halide perovskite memristors as a source of entropy. The paper proposes measurements and claims that the proposed architecture shows the features to create a strong PUF resilient to machine learning attacks. The work is well written and the Authors provide some interesting insight on a possible implementation of PUFs with perovskites. I have the following comments:

We thank the referee for the positive comments on the significance of our work. Our responses to the comments are as follows.

1) The Authors claim that the proposed 1D Halide Perovskite based memristors show endurance to the write cycles, however it is not clear whether the changes over the several cycles would actually modify the expected CRP, a discussion on this aspect would be useful in particular referring to the proposed write back scheme.

We understand the reviewer's concern and thank him/her for this comment. The write cycles of memristors in general (including the most popular oxide materials such as TiO_x , TaO_x and HfO_x) portray high cycle-to-cycle variations [Lastras-Montaña, M.A. et al. 2018. *Nature Electronics*, 1(8), pp.466-472.; Lee, J.H. et al. 2019. *IEEE Transactions on Electron Devices*, 66(5), pp.2172-2178.]. Our perovskite devices are no less susceptible to such endurance variations. Hence with each write cycle, we expect the CRP space to be modified. We in fact utilize this feature to our advantage- to physically reconfigure our memPUFs as explained below. We tackle this using 2 strategies:

1. We use only 1 write operation (during the enrolment phase) to generate the CRPs. In the deployment phase, the PUFs are only read with a small reading voltage (e.g. 0.1 V) to verify the bits. This is in contrast to some previous proposals that generate bits based on memory writes, e.g.

- a. Rose, G. S. & Meade, C. A. Performance analysis of a memristive crossbar PUF design. in 2015 52nd ACM/EDAC/IEEE Design Automation Conference (DAC) 1–6 (IEEE, 2015)
- b. Yang, J. et al. A physically unclonable function with BER < 0.35% for secure chip authentication using write speed variation of RRAM. in 2018 48th European Solid-State Device Research Conference (ESSDERC) 54–57 (IEEE, 2018).

We had mentioned this in the earlier version of the supporting information. But because of the reviewer's concern, we have now moved this statement to the revised main text. Please refer to page 8 lines 26-29: *"Note that unlike some previous proposals that generate bits based on memory writes, our method only performs write once and hence can be used to uniquely identify a device for authentication without suffering from the limited write endurance of 1-D HP RRAM devices."*

2. We utilize these cycle-to-cycle variations to our advantage- to physically RE-configure our memPUF array to support ownership change, change in privileges (e.g. software version downgrade) or prevent information leakage due to overuse. The experimental measurements of reconfiguration and analysis of the PUF metrics such as the reconfiguration flowchart, analog color and digitized checkerboard maps before and after reconfiguration, histogram of the reconfiguration HD using 10 cycles of consecutive writes and the correlation matrix is shown in Fig. 4 of the main text. With physical reconfigurability, the trustworthy party can now reconfigure all keys, making the previous keys inferred and documented by an attacker useless. Moreover, as opposed to logical reconfiguration which require additional circuits, this approach saves area and power overheads.

Please refer to the abstract line 11. We have also briefly mentioned this when introducing the memristor characteristics. Please refer to page 6 lines 23-24- *"The cycle-to-cycle variations are exploited for reconfiguration, as discussed later."*

2) The Authors report that their PUF consists of a 32×32 array of RRAM dot-point devices as shown in Fig. 3a which they logically treat as a crossbar, leaving to future work to physically implement an actual crossbar, how does this assumption impact their analysis and the reported results?

We thank the reviewer for this comment. We expect all our findings to remain valid even with future implementations of large crossbar arrays of HP memristors. The only scenario where our experimental measurements on dot point devices could differ from crossbar implementations would be when the resistance of the contacts lines in the crossbar interferes with our measurements of the high (HRS) and low resistance states (LRS). To determine this, a small 8×8 crossbar array of HP memPUFs was fabricated and the contact resistance of the lines (thickness = 50 nm, width= 200 μm and length: 17.5 mm) were extracted. The resistance of the top and bottom electrodes was found to be equal to 65 ohms, far lower than our LRS of $\sim 350\text{-}400$ ohms (Figure R1a, this is now inserted as Supplementary Figure 7a). Further, dividing this by 8 results in an unit cell resistance of the wiring to be only ~ 8 ohms. Hence, we expect our results and approach to hold well with future implementations of large crossbar arrays of halide perovskite memristors. We also show below representative IV characteristics of a dot point and crossbar HP memristor with the same device area (Figures R1b-c, this is now inserted as Supplementary Figures 7b-c). Both configurations yield similar results per se. However, we are currently limited by the poor yield of large crossbar arrays of halide perovskite memristors with reasonable endurance and hence, we implement dot point arrays for this work. With focussed engineering efforts to improve the yield of perovskite crossbar memristors, we expect to experimentally fabricate large-scale halide perovskite PUF chips in the near future.

Figure R1. **a** Contact line resistance of the top and bottom electrodes of a 8×8 crossbar array of HP memristors. Representative IV characteristics of a **b** dot point and **c** crossbar HP memristor with the same device area.

We have now added this discussion as Supplementary Note 12 and have referred to this in the main text (please refer to page 17 lines 1-3).

3) When analyzing the ML attack resilience, the Authors report figure 5.1-b that their proposed approach based on recurrence lowers the accuracy of a ML attack to 52% in the worst case, however the trend of the curves does show a marked increase with the increase of training samples. This induces to infer that a higher number of samples could increase significantly the accuracy of the attack, the Authors should comment on this aspect.

We thank the reviewer for the comment. In the earlier submitted version, we had presented ML attack accuracies after training with 10^5 samples (Figs. 5 b, d). To determine the limits of our system and to analyse the extend of this increase, we have now performed new experiments on ML attack accuracies with 10x increased number of samples, i.e. 10^6 samples. The data is presented below in Figure R2.

Firstly, it is noted that the trend of better attack resilience with recurrence still holds true with higher number of CRPs (Figure R2 b vs a). Secondly, it can be seen from Figure R2b that the rate of increase of ML attack accuracy drops sharply on increasing the number of samples and is expected to saturate at values $< 60\%$.

Although Figure R2b shows the accuracies increasing with number of CRPs for the Bagged Trees and Adaboost algorithms, the rate of increase is approximately 0.7 % for every doubling of training CRPs even after collecting around 1.6 million CRPs. For the worst case assuming constant slope and extrapolating from 2^{20} CRPs as shown in Figure R2c, for which the accuracy is 54 %, the attacker would need to collect 2^{88} CRPs to increase the accuracy by 46 % to 100 %. With an accuracy of 100 %, the PUF has been successfully impersonated by the attacker because the accuracy has increased beyond the reliability of our PUF [Lim, Daihyun, et al. "Extracting secret keys from integrated circuits." *IEEE Transactions on Very Large Scale Integration (VLSI) Systems* 13.10 (2005): 1200-1205]. However, the rate at which CRPs are collected do not typically exceed 2Mbps [Chip Hong Chang et al. "A low power diode-clamped inverter-based strong physical unclonable function for robust and lightweight authentication." *IEEE Transactions on Circuits and Systems I: Regular Papers* 65.11 (2018): 3864-3873]. Nevertheless, even if we assume an aggressive 10Mbps CRP collection rate, it will take 9.8×10^{11} years for the attacker to collect 2^{88} CRPs and achieve close to 100 % accuracy, demonstrating that our HP memPUFs are secure against the aforementioned ML attacks. We have now added this analysis as Supplementary Figure 6 in Supplementary Note 8. Please refer to the Pages 15-16 in the Supporting Information and Page 14 in the main text).

Figure R2. 1-D HP Strong memPUF resilient to Machine Learning Attacks. **a** Machine learning (ML) results for strong PUF without recurrence upon training with 10^6 samples. Accuracies close to 90 % reveals the HP memPUF to be highly susceptible to such attacks. **b** ML results for strong PUF with recurrence upon training with 10^6 samples. Accuracies reduce to almost wild guess probability with recurrence compared to **a**, proving that recurrence improves resistance to ML attacks. **c** Extrapolation of the attack accuracy for very large number of training samples. Plot shows accuracy reaching almost 100 % for 2^{88} CRPs, however collecting these many CRPs in a reasonable amount of time is not feasible.

Reviewer #2 (Remarks to the Author):

The study demonstrates a large scale implementation of security primitives exploiting the switching physics of 1D halide perovskite memristors for authentication that is reconfigurable. They implemented a 1 kb memory structure for a weak PUF demonstration. The authors based their ideas on the co-existence and coupling of ionic and electronic components. The authors claim their work is "the very first implementation of a HP memPUF" hence it is strange to also claim "the largest ever implemented HP memristor array till date" as if it is the first on HP then it is for sure the largest! The material description, PUF analysis and outcome in my view are well argued and well presented. The type of material used in this paper gives me hope to see a much reduced cost in production of PUFs.

We thank the reviewer for commending our contribution and pointing out the significance of our experimental demonstration of PUFs with solution-processed halide perovskites. We also very much appreciate the reviewer's suggestions to improve our manuscript by making more objective statements.

We say "the largest ever implemented HP memristor array till date" strictly from a memristor implementation perspective. To the best of our knowledge, the largest halide perovskite memristor arrays implemented till date are reported in

1. Seo, J.Y. et al. 2017. Wafer-scale reliable switching memory based on 2-dimensional layered organic–inorganic halide perovskite. *Nanoscale*, 9(40), pp.15278-15285. Here halide perovskite films were deposited on a 4 inch wafer, but only 20 devices were measured to test for reliability.
2. Hwang, B. and Lee, J.S., 2017. A strategy to design high-density nanoscale devices utilizing vapor deposition of metal halide perovskite materials. *Advanced Materials*, 29(29), p.1701048. Here a 16x16 crossbar array was realized, but again only 10 devices were tested for reliability.

In this work, we fabricate flexible halide perovskite memristors and characterize 1kb or 1024 elements for the PUF analysis. Hence, we make this claim. To the best of our knowledge, we are also the first to report the experimental demonstration of a PUF implementation with halide perovskite memristors.

I am disappointed and it is unfortunate that this great work has to be rejected as it postpones one of the main aspects of their research product for future research. The authors states "However, the current 1D HP memPUF is still susceptible to large BERs with high temperature changes and is one of the major future directions of research." The REL calculation for intra-HD is not assessed the temperature, based on what I found. Could you explain what do you mean by putting "We calculate both temporal and voltage-induced reliability" right after stating "response at any other temperature/voltage to determine if the responses are different (unreliable) or not (reliable)" in the Supp.? Are you linking temporal with temperature or are you implying the impact is transient? The Reliability description in this paper is very unclear and vague in my view. I suggest authors to at the very least demonstrate that devices respond on average the same way to changes in temperature.

We thank the reviewer for the comment. We would like to first clarify that the temporal variations we mention in the manuscript refer to only transient variations with time. We do not assume temperature effects when we refer to "temporal" reliability. Temporal reliability is critical to be analysed from a PUF perspective since this determines the stability of the generated CRPs with time [Gassend, Blaise, et al. "Identification and authentication of integrated circuits." *Concurrency and Computation: Practice and Experience* 16.11 (2004): 1077-1098.]. Since memristors are vulnerable to read variations over time, this analysis becomes even more critical. In this work, after we generate the CRP space in the enrolment phase, we experimentally measure the temporal stability of each of the 1024 devices of the HP memPUF array over 100 read cycles. The BER or intra-HD values presented in Supplementary Figure 5a presents the Bit-error rates (BER) due to temporal noise only.

To evaluate the robustness of our PUFs to noisy voltage channels, we analyse our memPUF array with a higher voltage that is 1.5x (50 % change) the normal reading voltage. This is also in line with the standard test protocols adopted in literature [Gassend, Blaise, et al. "Identification and authentication of integrated circuits." *Concurrency and Computation: Practice and Experience* 16.11 (2004): 1077-1098.]. In both these cases, our HP memPUF array is observed to be vulnerable to bit flips or errors, similar to other state of the art technologies. BERs of 2.33 % and 19.25 % are obtained when analysed against temporal-only (Supplementary Figure 5a) and voltage + temporal (Supplementary Figure 5b) fluctuations respectively without write back. However, the write back strategy that we adopt utilizes the widely separated bimodal resistance profiles of the HP memristors to reliably set the devices to their complimentary states, enabling reliable generation of bits. The BER values reduce to effectively 0 % for both temporal and voltage noise conditions as illustrated in Supplementary Figures 5a-b. This approach allows us to overcome bit-flip errors without fuzzy extractors or majority voting, hence saving area and power overheads.

We understand the reviewer's concern regarding the temperature stability of these devices and also realize that this is one of the benchmarking metrics for a PUF. To evaluate our HP memPUF array against temperature variations, we have conducted new experiments with heating. The devices were heated to 60°C and the data was collected at this constant thermal stress. The experimental measurements at room temperature were compared to the measurements under the 60°C stress and the reliability of the devices were analysed without and with the write back step. At each temperature, the measurements are repeated 100 times and hence, these results naturally capture the effects of both temperature and temporal fluctuations. A checker board pattern of 16 representative devices is presented below as Figure R3, along with the associated bit error rate (BER) values. We observe that our HP memPUFs are vulnerable to thermal stress and the extracted BER values are as high as 63.42 % without write back. However, once again the write back strategy we adopt allows us to maintain the integrity of the generated bits and CRP space at temperatures as high as 60°C by exploiting the widely separated bimodal resistance profiles of the HP memristors (by reliably setting the devices to their complimentary states). We have now inserted this data to Supplementary Notes 5 and 6, Supplementary Figures 4 and 5 and made necessary changes to the main text (Please refer to page 11).

Figure R3. Reliability results with temperature and temporal fluctuations for 16 memristors. a shows the initial checkerboard pattern obtained from 16 devices. Out of 100 measurements taken over time, the

checkerboard patterns in **a**, **b**, **c** are shown for one representative measurement. After applying a thermal stress of 60°C, checkerboard pattern in **b** shows 4 bits flipping without write-back compared with **a**, and no bits flipping with write-back when **c** is compared with **a**. BER color map shows errors across temperature variations in **d** without write-back and no errors in **e** with write-back. 100 consecutive reads are done for both **d** and **e**.

REVIEWER COMMENTS

Reviewer #1 (Remarks to the Author):

I would like to thank the Authors for revising their paper, some of my questions have been answered, however I still am not convinced about the endurance analysis. The Authors claim that their PUF is written only during enrollment phase leaving further writes to the possible reconfiguration of the device. However the Authors also discuss that their proposed methodology to improve the reliability of their PUF is to use a write back approach, therefore this seems in contrast with their answer related to endurance. Don't these write backs affect the endurance? And if this is not the case the Authors should clarify it in the text.

Reviewer #2 (Remarks to the Author):

My thanks to the authors for making the relevant changes required. While I am reasonably okay with all responses, I am not satisfied entirely with one point in particular. The temperature range typical to semiconductor use in IC design traditionally is ranging from -55°C to $+125^{\circ}\text{C}$ and this is not considered harsh or extreme temperatures for an IC. Of course beyond this window, extreme requirements kick in and particular applications are in discussion. Other traditional ranges are up to $+85^{\circ}\text{C}$ and some as low as $+70^{\circ}\text{C}$. The choice of 60°C is a bit weird and random unless there is an explanation to it.

REVIEWER COMMENTS

Reviewer #1 (Remarks to the Author):

I would like to thank the Authors for revising their paper, some of my questions have been answered, however I still am not convinced about the endurance analysis. The Authors claim that their PUF is written only during enrollment phase leaving further writes to the possible reconfiguration of the device. However the Authors also discuss that their proposed methodology to improve the reliability of their PUF is to use a write back approach, therefore this seems in contrast with their answer related to endurance. Don't these write backs affect the endurance? And if this is not the case the Authors should clarify it in the text.

We thank the referee for the positive comments on our work and the first round of revision. We understand the reviewer's concern and thank him/her for this comment. To clarify the concern regarding endurance, write back and reconfiguration, we would like to reiterate the procedure we propose to deploy the perovskite PUFs.

1. During the enrolment phase, devices in the HRS are “written” to their LRS and then “erased” back to the HRS. This constitutes 1.5 cycles of endurance. Stochastic variations in the HRS between the PUF cells is used to generate a decision matrix, to “write-back” certain cells to their LRS. This corresponds to 2 cycles of endurance for a memristor PUF cell. After “write back”, a digital map of the CRP space is “read” out using a small reading voltage = 0.1 V.

2. When the PUF is deployed in the field, “read” operations are used to verify the CRPs and identify and authenticate the PUF array. Since the “read voltages” used are very small ($\ll V_{\text{set}}$ of the memristor cells), we do not expect to encounter false “write” operations that may unintentionally change the CRPs. We show that our devices are robust to temporal, voltage and temperature fluctuations during this entire procedure, attesting the robustness and significance of our approach (Supplementary Figs. 4 and 5).

3. When reconfiguration is deemed necessary such as to support ownership change, change in privileges (e.g. software version downgrade) or prevent information leakage due to overuse; we utilize the cycle-to-cycle variations in the memristor's switching behaviour, to reconfigure the CRP space. In this stage, the PUF cells would be “erased” back to their HRS and steps 1 and 2 would be repeated. Each reconfiguration step constitutes 1 endurance cycle. Given the average endurance of our devices is ~ 450 cycles (Fig. 2f), in principle we could reconfigure our PUF cells $450-2 = 448$ times. In Fig. 4, we show that such reconfiguration results in a renewed CRP space with good independence of the key bits and low correlation to the previous CRP space, highlighting the significance and utility of our perovskite memristor PUFs, and the approach we undertake to enrol and deploy such devices on field.

Hence, write back does not take a major toll on the endurance of our devices and the cycle-to-cycle variations can be smartly exploited to create reconfigurable PUFs.

Reviewer #2 (Remarks to the Author):

My thanks to the authors for making the relevant changes required. While I am reasonably okay with all responses, I am not satisfied entirely with one point in particular. The temperature range typical to semiconductor use in IC design traditionally is ranging from -55°C to $+125^{\circ}\text{C}$ and this is not considered harsh or extreme temperatures for an IC. Of course beyond this window, extreme requirements kick in and particular applications are in discussion. Other traditional ranges are up to $+85^{\circ}\text{C}$ and some as low as $+70^{\circ}\text{C}$. The choice of 60°C is a bit weird and random unless there is an explanation to it.

We thank the referee for the positive comments on our work and the first round of revision. We also very much appreciate the reviewer's suggestion to test our PUFs at temperatures that are relevant to the electronics industry. We believe this has allowed us to improve our manuscript as well as the practical

relevance of our work. We apologize for not having conducted the first round of experiments in accordance with the industry grade benchmark. We have now conducted additional experiments to verify the robustness of our PUFs to temperatures as high as 85°C. The results are presented below.

To evaluate our HP memPUF array against temperature variations, we have conducted new experiments with heating. The devices were heated to 85°C and the data was collected at this constant thermal stress. The experimental measurements at room temperature were compared to the measurements under the 85°C stress and the reliability of the devices were analysed without and with the write back step. At each temperature, the measurements are repeated 100 times and hence, these results naturally capture the effects of both temperature and temporal fluctuations. A checker board pattern of 16 representative devices is presented below as Fig. R4, along with the associated bit error rate (BER) values. The extracted BER values are as high as 63.71% without write back Fig. R5. However, once again the write back strategy we adopt allows us to maintain the integrity of the generated bits and CRP space at temperatures as high as 85°C by exploiting the widely separated bimodal resistance profiles of the HP memristors (by reliably setting the devices to their complimentary states). We have now added this new data measured at 85°C to Supplementary Notes 5 and 6, Supplementary Figures 4 and 5 (Please refer to pages 11-14). The new results are also shown below as Fig. R4 and R5.

Figure R4. Reliability results with temperature and temporal fluctuations for 16 memristors. a shows the initial checkerboard pattern obtained from 16 devices. Out of 100 measurements taken over time, the checkerboard patterns in a, b, c are shown for one representative measurement. After applying a thermal stress of 85°C, checkerboard pattern in b shows 3 bits flipping without write-back compared with a, and no bits flipping with write-back when c is compared with a. BER color map shows errors across temperature variations in d without write-back and no errors in e with write-back. 100 consecutive reads are done for both d and e.

Figure R5. Bit-error rates (BER) due to **a** temporal noise obtained by performing 100 consecutive reads, **b** 50 % change in read voltage, and temperature fluctuations of **c** 60°C and **d** 85°C. Improvements in BER by using Temporal Majority Voting and Write-Back (WB) are shown for all cases. WB reduces BER to 0 % and outperforms other TMV schemes.

REVIEWERS' COMMENTS

Reviewer #1 (Remarks to the Author):

I have no further comments.

Reviewer #2 (Remarks to the Author):

I would like to thank you for providing a sufficiently satisfying response on write back. I am satisfied and endorse the publication.

REVIEWER COMMENTS

Reviewer #1 (Remarks to the Author):

I have no further comments.

We thank the referee for the positive comments on our work and the revised changes.

Reviewer #2 (Remarks to the Author):

I would like to thank you for providing a sufficiently satisfying response on write back. I am satisfied and endorse the publication.

We thank the referee for approving the revised changes and endorsing our work.